# Enhancement of Carrier Migration by Monolayer MXene Structure in Ti$_3$CN/TiO$_2$ Heterojunction to Achieve Efficient Photothermal Synergistic Transformation of CO$_2$

Chenxuan Zhu [1,2], Mingnv Guo [3,*], Ziqi Wang [1,2], Jiang He [1,2], Jiaqi Qiu [1,2], Yuxuan Guo [3], Yunfei Yan [1,2], Jingyu Ran [1,2] and Zhongqing Yang [1,2,*]

1. Key Laboratory of Low-Grade Energy Utilization Technologies and Systems, Chongqing University, Ministry of Education, Chongqing 400044, China
2. School of Energy and Power Engineering, Chongqing University, Chongqing 400044, China
3. School of Mechanical and Power Engineering, Chongqing University of Science and Technology, Chongqing 401331, China
* Correspondence: guomingnv@cqust.edu.cn (M.G.); zqyang@cqu.edu.cn (Z.Y.); Tel.: +86-023-65022338 (M.G.); +86-023-65103115 (Z.Y.)

**Abstract:** Carbon nitride MXene exhibits good metal conductivity, high photothermal conversion, carrier mobility, and high exposure of active sites, which makes it a promising co-catalyst for photothermal synergistic transformation of CO$_2$. In this paper, Ti$_3$CN/TiO$_2$ heterojunction was constructed in situ using Ti$_3$CN as TiO$_2$ precursor to investigate the performance of Ti$_3$CN MXene in photothermal synergistic transformation of CO$_2$, and then the monolayer structure was utilized to enhance the interfacial charge transfer and improve the photothermal catalytic activity of Ti$_3$CN. The catalysts were characterized by SEM, XRD, XPS, and UV-Vis DRS, and it was found the heterojunction constructed by monolayer MXene had a narrower bandgap and a higher carrier generation mobility, which, combined with the catalytic activity test, proved the single monolayer Ti$_3$CN MXene had better photothermal synergistic conversion efficiency of CO$_2$, and the heterojunction yield was 11.36 μmol·g$^{-1}$·h$^{-1}$ after layering, compared with that before layering (9.41%), which was 1.2 times higher than that before layering (9.41 μmol·g$^{-1}$·h$^{-1}$).

**Keywords:** photothermal synergistic; CO$_2$ reduction; MXene-based heterojunction

## 1. Introduction

Reducing CO$_2$ to high-value fuel through photothermal synergistic catalysis and converting solar energy with low unstable energy density into chemical energy in a stable, high-energy density, and high value-added fuel is an important means to realize the resource utilization of CO$_2$ and to deal with global warming and the world energy crisis [1]. Photothermal synergistic catalysis uses ultraviolet-Visible light (UV-Vis) to excite photogenerated electron hole pairs in semiconductor catalysts, and electrons transition from the valence band to the conduction band, which is used to activate CO$_2$ molecules and reduce the excessive reaction energy barrier. At the same time, absorbing Visible-infrared light (Vis-IR) to increase the surface temperature of the catalyst, using heat energy to promote the separation and migration of photogenerated carriers, improves the ability of light absorption and carrier transport, resulting in the synergistic effect of light and heat [2]. Compared with the photoelectric catalysis mode, photothermal synergistic catalysis only needs a single energy input of sunlight, which does not cause excess energy consumption. Through the qualitative utilization of solar energy, high solar energy absorption and conversion rate can be achieved.

TiO$_2$ is the most suitable catalyst for large-scale industrial application because of its excellent charge transfer quality, stable chemical composition, low cost, and no pollution.

However, $TiO_2$ cannot respond to Visible-infrared light, and its ability for photothermal conversion is not good, and thus sunlight is not fully utilized. Precious metals are often introduced as co-catalysts to expand the range of light response, improve photothermal conversion, and increase carrier mobility [3,4], but the introduction of precious metal co-catalysts increases the cost and the possibility of pollution.

MXenes have become a potential substitute for precious metal cocatalysts [1]. The general formula of MXenes is $M_{n+1}AX_nT_x$, where M represents transition metal elements, A is from IIIA and IVA elements, X is N or C elements, and T represents surface terminal groups (-OH, -F, etc.) [5]. MXenes refer to two-dimensional transition metal carbides, carbonitrides, and nitride families. As a new type of two-dimensional materials with a graphene-like structure, MXene exhibits good metal conductivity, high photo-thermal conversion, carrier mobility, and tunable electronic structure of surface end groups [6], which is suitable for use as a co-catalyst in photothermal catalytic material systems to promote the separation and migration of photogenerated carriers and to provide effective active sites for the reaction [1,7]. By introducing MXene co-catalysts to construct MXene-based composite photocatalysts, the problems of low light absorption and utilization efficiency, low photothermal conversion, and easy recombination of photogenerated carriers in the process of photothermal synergistic catalytic reaction can be further improved, which is of great significance for improving photocatalytic activity. Researchers in the field of photothermal synergistic catalysis have noticed the advantages of MXene-based heterojunctions. And, research on this catalyst is continuing [1,8].

When the M element is Ti, MXene can provide a natural platform for the construction of $TiO_2$. Titanium atoms on MXene may become the nucleation center of the $TiO_2$ photocatalyst, and a close interface connection is formed between MXene and $TiO_2$ photocatalyst, to minimize the recombination of photogenerated carriers induced by defects [9]. So far, a large number of $MXene/TiO_2$ composites have been constructed for the conversion of $CO_2$ [10–13]. It is proved this design can effectively broaden the light absorption range of $TiO_2$ and improve the migration efficiency of photogenerated carriers in $TiO_2$, which is obviously beneficial to the conversion of $CO_2$. The structure of $Ti_3CN$, in which nitrogen atoms randomly replace carbon atoms and the titanium layer is stacked in a hexagonal structure, is very similar to that of $Ti_3C_2$ [14]. The lone pair of electrons of reactive nitrogen in $Ti_3CN$ gives it a stronger electrical conductivity than that of the carbide MXene [14,15]. What is more, $Ti_3CN$ shows a higher optical absorption rate in the near-infrared (NIR) region compared to that of $Ti_3C_2$ [15]. $Ti_3CN$ exhibits higher light absorption in the near infrared region compared to $Ti_3C_2$. However, there is a lack of attention to $Ti_3CN$ MXene, which has a stronger photothermal conversion ability and higher carrier transport efficiency [16,17].

The structure of low-dimensional materials will affect their electronic properties to a great extent. The study of layered nanostructures is of great significance in the field of $CO_2$ transformation [6,18]. It has been proved by simulation calculation the layer thickness of MXene has a significant effect on its catalytic performance [19]. But, the research on MXene-based heterojunctions mainly focuses on the selection of MXene materials or the construction methods of heterojunctions, and the influence of the intrinsic structure of MXene on the multi-catalytic performance needs to be explored in depth [20]. After etching, MXene has a large-size accordion-like layered bulk structure, and the large size is unfavorable for carrier migration between the MXene and semiconductor interfaces, and the air-filled gaps between the layers reduce the conductivity and further hinder the carrier transfer [21]. Therefore, it is important to explore the performance of $Ti_3CN$ MXene in the field of photothermal synergistic transformation of $CO_2$, the enhancement of the photothermal catalytic performance of the catalysts by decreasing the size and the number of layers of MXene, and the design of reasonably designed Mxene-based composite photocatalytic materials which have a positive effect on promoting the further application of two-dimensional MXene for photothermal synergistic catalysis.

## 2. Results and Discussioon

### 2.1. Analysis of Catalyst Morphology and Structure

2.1.1. SEM

Using emission scanning electron microscopy (SEM) to characterize the micro-morphology of the samples, the results are shown in Figure 1. Figure 1a,b show the MXene carriers with different morphologies, which are multilayered $Ti_3CN$ versus monolayer $Ti_3CN$, respectively. Figure 1a shows the un-sonicated multilayered $Ti_3CN$, the mTCN is in the form of a layered block structure, which is in accordance with the "accordion-like" morphology of MXene, attributed to the removal of Al from the blocky $Ti_3AlCN$ MAX phase precursor. Figure 1b demonstrates the microscopic morphology of single/few layers of $Ti_3CN$ separated by differential centrifugation after ultrasonic layering treatment; compared with multilayered $Ti_3CN$, there is a clear separation between the layers of dTCN, which shows a single and few layers of lamellar structure. This is due to the fact that the bulk structure of multilayer $Ti_3CN$ is disrupted during the ultrasonication process, and the layers connected by van der Waals forces are gradually separated from each other. Meanwhile, during the subsequent separation of single/few-layer MXene using differential centrifugation, water molecules as well as residual $Li^+$ would further intercalate the MXene, resulting in further expansion of the interlayer distance [22]. The ultrasonically treated MXene has a sub-micron transverse size [23,24] and the width is concentrated between 200–500 nm [24]. Therefore, dTCN has smaller size, fewer layers, and larger interlayer distance than mTCN.

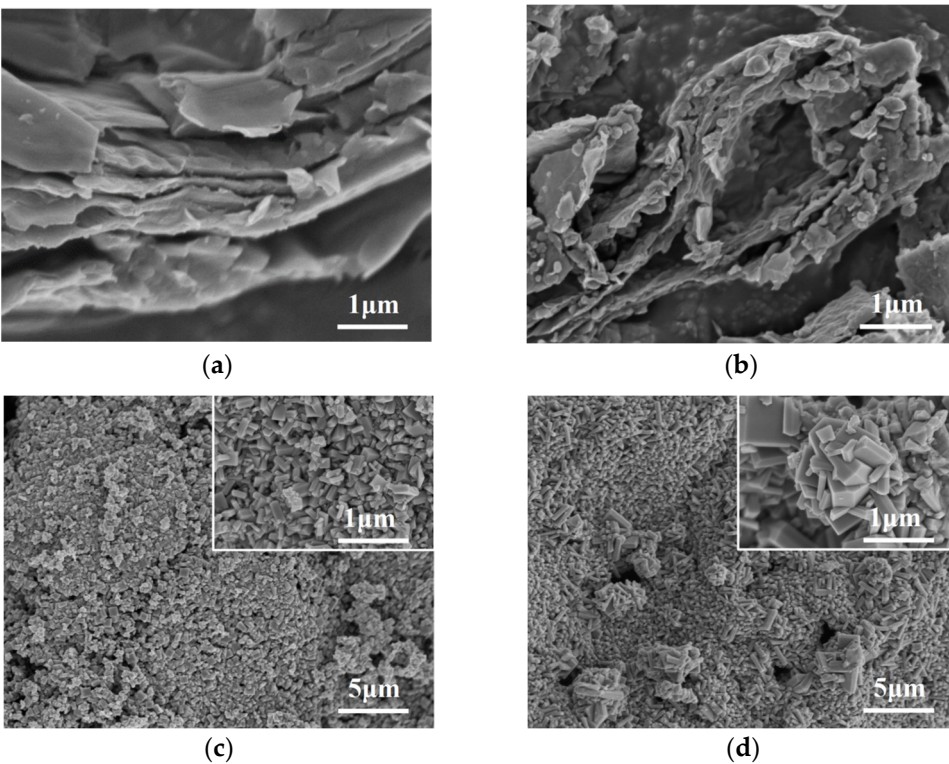

**Figure 1.** SEM images of $Ti_3CN$ and the $Ti_3CN/TiO_2$ heterojunction constructed in situ on it: (**a**) mTCN; (**b**) dTCN; (**c**) mTOCN; and (**d**) dTOCN.

Figure 1c,d shows the $Ti_3CN/TiO_2$ heterojunction obtained by hydrothermalization of multilayer $Ti_3CN$ and single/few-layer $Ti_3CN$ in acidic hydrothermal fluid at 180 °C for 4 h, respectively. Under hydrothermal conditions, $Ti_3CN$ acts as a Ti source and nucleation center, using Ti atoms to achieve interfacial bonding and ensure the subsequent efficient photogenerated charge diffusion. $TiO_2$ nucleates at the defects on the surface of $Ti_3CN$ and exposes the (101)-face under the effect of the morphology-directing agent $NaBF_4$ [9],

and the surface of Ti₃CN is homogeneously covered by the anatase TiO₂ at the end of the hydrothermal process.

To further explain the effect of ultrasonic delamination on the morphology of Ti₃CN, it was observed by atomic force microscope (AFM). The thickness of MXene nanosheets before and after delamination is shown in figure Figure 2a,b. The thickness of mTCN nanowires is 1~4 nm, and the thickness of dTCN is about 0.5~1 nm. AFM confirmed the successful synthesis of monolayer Ti₃CN. Ultrasound not only delaminated Ti₃CN but also reduced the transverse size of nanowire. The decrease in the thickness of nanowires can effectively reduce the migration distance of carriers on the materials, restrain carrier-hole recombination, prolong carrier lifetime, and improve the photothermal catalytic activity.

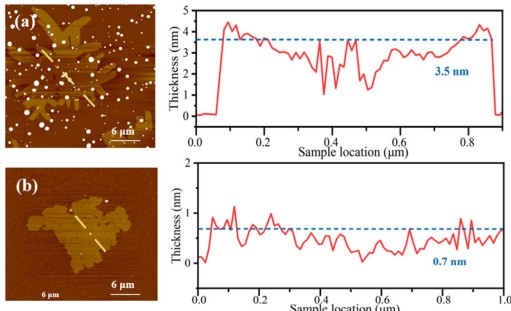

**Figure 2.** Representative AFM images and the corresponding height profiles of the yellow dotted line: (**a**) dTCN; and (**b**) mTCN.

### 2.1.2. XRD Crystal Structure Analysis

Figure 3 shows the X-ray diffraction (XRD) patterns of a monolayer Ti₃CN MXene, multilayer Ti₃CN MXene, and the Ti₃CN/TiO₂ heterojunction constructed in situ on them, which further shows the successful synthesis of Ti₃CN MXene carrier and Ti₃CN/TiO₂ heterojunction. Ti₃AlCN is etched by HF generated in situ by LiF and HCl, and the reflection peaks of mTOCN, dTOCN can be clearly observed at 6.6 and 7.2, respectively. Ti₃AlCN, the precursor of the MAX phase, gradually widens and disappears at 9.58°, 38.95°, indicating the removal of Al and the successful etching of Ti₃CN [25]. The crystallinity and crystal plane distance of monolayer MXene after ultrasound are lower than those of multilayer Ti₃CN. After hydrothermal oxidation, obvious anatase TiO₂ (JCPDS 84-1285) diffraction peaks were observed in two kinds of heterojunction samples, with some sharp diffraction peaks detected at 25.3°, 37.8°, 39.1°, 48.0°, 53.8°, 55.0° and 62.6°, which were attributed to (101), (004), (112), (200), (105), (211), (204) of anatase TiO₂, respectively. Ti₃CN has no obvious diffraction after hydrothermal treatment [22] which is attributed to the fact that the surface of Ti₃CN is covered by TiO₂ and is well distributed in heterojunctions [9]. In hydrothermal synthesis of TiO₂ under acidic conditions, the appropriate concentration of HCl and NaBF₄ can control the appearance of rutile TiO₂ in the catalyst.

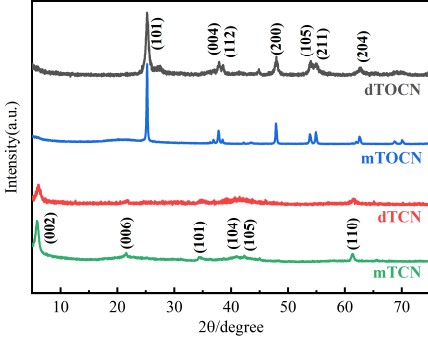

**Figure 3.** XRD spectra of dTOCN, mTOCN, dTCN, and mTOCN.

### 2.2. Performance Analysis of Photothermal Synergistic Transformation of $CO_2$

2.2.1. Analysis of Photothermal Synergistic Transformation of $CO_2$ Performance of mTOCN and dTOCN

The reduction product of $Ti_3CN$-based heterojunction photothermal catalysis of $CO_2$ is CO, as shown in Figure 4a, the average CO yield is 11.36 $\mu mol \cdot g^{-1} \cdot h^{-1}$ for dTOCN and 9.41 $\mu mol \cdot g^{-1} \cdot h^{-1}$ for mTOCN, which is 5.63 and 4.67 times higher than that of P25, respectively. MXene-based materials are regarded as plasmonic materials because of their very narrow bandgap and high conductivity [13]. MXene-based materials are considered metal-like materials with plasma properties due to their very narrow bandgap and high conductivity, and $TiO_2$ is an n-type semiconductor, and the metal-semiconductor structure composed of the two has a LSPR effect, which makes it possible to break through the limitations of the semiconductor bandgap width to realize the full-spectrum utilization of photothermal co-catalysis. When MXene is in close contact with the $TiO_2$ semiconductor, energy band bending occurs inside the semiconductor, resulting in a Fermi energy level equilibrium between MXene and $TiO_2$ and the establishment of an ohmic junction [13]. This implies $Ti_3CN$ tends to continuously inject electrons into $TiO_2$ to maintain charge equilibrium, which will significantly increase the carrier mobility and promote the photothermal catalytic activity of the $Ti_3C_2/TiO_2$ heterostructures.

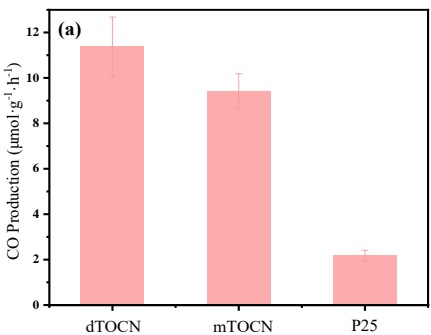 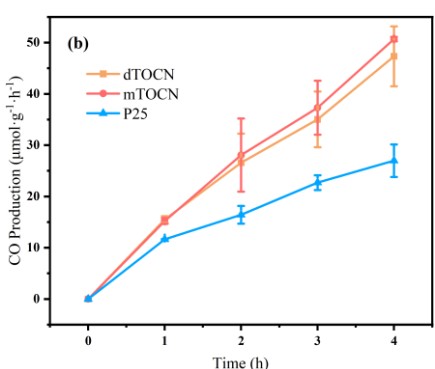

**Figure 4.** The catalytic testing of the dTOC, mTOCN, and P25 in photothermal synergistic catalytic $CO_2$ reduction. One can see that the main reduction products was CO: (**a**) the Photothermal synergistic catalytic activity of dTOC, mTOCN, and P25; and (**b**) the rate of product formation over time.

The catalytic reaction occurs when $TiO_2$ is photoexcited to produce photogenerated electron-hole pairs, while at the same time hot electrons are excited from $Ti_3CN$. The electrons generated by $Ti_3CN$ are transferred to the CB of the titanium dioxide semiconductor. Because the work function of $Ti_3CN$ is lower than of $TiO_2$ [26], the electrons cannot flow back toward $Ti_3CN$. In this regard, the spatial separation of photogenerated electron-hole pairs is realized. The efficiency of interfacial charge transfer is heavily dependent on the contact interface, and in MXene-based heterojunctions, good charge transfer is facilitated by the in situ growth of heterojunctions on $Ti_3CN$ due to $TiO_2$ without additional carrier diffusion to maintain good contact.

The catalytic efficiency of dTOCN is significantly higher than that of mTOCN. The delaminated $Ti_3CN$ not only has a higher carrier separation and migration efficiency due to smaller structural dimensions, but also exposes more active surfaces, and thus has the following advantages: (1) because the small size of monolayer $Ti_3CN$ has rather short charge transfer distance of the ultrathin structure, the photogenerated holes from the $TiO_2$ conduction band (CB) are efficiently transferred and accumulate on the $Ti_3CN$ surface across the heterojunction. (2) The specific surface area of monolayer $Ti_3CN$ increases and provides a better growth platform for $TiO_2$, so dTOCN has more abundant heterojunction interface connections.

### 2.2.2. Analysis of Photothermal Synergistic Effect of dTOCN

Catalyzed $CO_2$ reduction was tested to explore the photothermal synergistic effect of dTOCN with better catalytic effect. By adjusting the wavelength of the initiating light and the temperature inside the reactor to change the form of energy input to the reaction system, the full-band light, UV-Visible band, and infrared band corresponded to the realization of photothermal, pure light, and pure thermal catalytic conditions, respectively. The $CO_2$ reduction performance of dTOCN under the three energy inputs was then tested.

Figure 5 shows the average CO yield of dTOCN under different catalytic conditions within 4 h. The thermal energy provided by IR light alone is not enough to make the reaction cross the activation energy barrier, and the reduction product CO could not be detected in the reactor. The light energy provided by UV-Visible light excites the photogenerated carriers inside the heterojunction and the $CO_2$ is activated by the electrons migrating to the surface of the catalyst, and the reduction product CO can be detected in the reactor with an average precipitation rate of 7.51 $\mu mol \cdot g^{-1} \cdot h^{-1}$. Whereas, under the full-spectrum illumination, the UV-Visible waves generate excitation photogenerated carriers to produce photochemistry, and the Visible-infrared part excites thermal energy to produce photothermal effect, and the CO precipitation rate increases to 11.15 $\mu mol \cdot g^{-1} \cdot h^{-1}$ under the synergistic effect of the two, and the catalytic activity is greatly improved [1]. It can be seen for the $Ti_3CN/TiO_2$ heterojunction, photothermal catalysis is not a simple superposition of photocatalysis and thermal catalysis, and there is a strong synergistic effect between light and heat [27]. On the one hand, UV-Visible illumination excites the carriers of the catalysts, and the transport of carriers facilitates $CO_2$ activation, and on the other hand, the Visible-infrared portion provides the thermal energy for the reaction system, and the thermal energy, in the meantime of degrading the inverse activation energy, further promotes the separation of carriers, which leads to a significant improvement of the catalytic activity [27]. The thermal energy further promotes the separation of carriers while reducing the activation energy, thus realizing the photothermal synergistic and efficient reduction of $CO_2$ in the full spectrum.

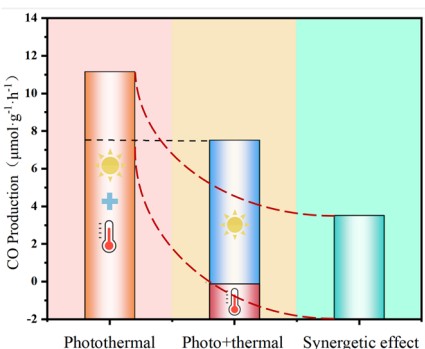

**Figure 5.** Catalytic activity and photothermal synergistic effect of dTOCN under different catalytic conditions.

The internal temperature of the catalytic reactor was adjusted by changing the circulating cooling water temperature outside the reactor, and the cumulative precipitation rates of $CO_2$ reduction products of dTOCN photothermal catalytic CO reduction over time under the conditions of 10 °C, 20 °C, and 30 °C were measured, and the results were shown in Figure 6, which showed the precipitation rate of the target products of CO was positively correlated with the reaction temperature, and the thermal energy played a promotional role for the catalytic activity.

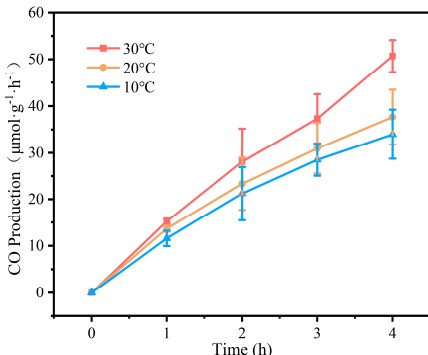

**Figure 6.** Photothermal catalytic yield of dTOCN with time at different temperatures.

The stability of dTOCN and mTOCN is tested under photothermal conditions, and the test results are shown in the Figure 7. The reaction was carried out for 4 h under photothermal conditions, then the air was extracted from the reactor by vacuum pump, and the mixture of $CO_2$ and $H_2O$ was introduced into the reactor by bubbling. After a period of time, the mixture of $CO_2$ and $H_2O$ was filled with the reactor for five times, and then the spiral ports on both sides of the reactor were closed for the next round of reaction for four times. The stability test results show the catalyst has good stability.

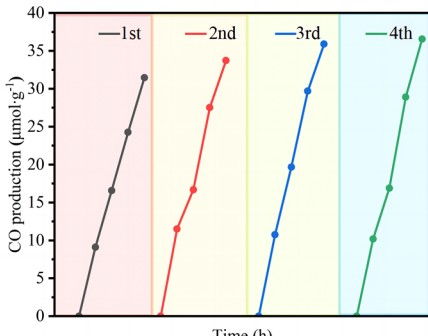

**Figure 7.** Photothermal catalyst lifespan tests over dTOCN. The testing time is divided into four cycles, with each cycle lasting 4 h. The different color of the line represents the number of cycle.

### 2.3. Analysis of Chemical Components and Photoelectrochemical Properties of mTOCN and dTOCN

#### 2.3.1. XPS Chemical Component Analysis

X-ray photoelectron spectroscopy (XPS) analysis was performed to elucidate the chemical elemental composition of the prepared sample surfaces and the bonding state of each element. All binding energies were calibrated by the C1s peak located at 284.8 eV, which belonged to the indefinite carbon (C-C). The full XPS spectra of mTOCN and dTOCN are shown in Figure 8, each vibrational peak in the spectra is labeled, from which it can be seen the heterogeneity contains the elements of C, N, O, Ti, and F. The element of F is derived from the surface end-groups -F on the surface of the $Ti_3CN$ MXene after etching with LiF and HCl, and the element of O belongs to the surface end-groups -O, -OH, and -H of the $TiO_2$ and $Ti_3CN$ MXene, and part of the Ti and part of the O elements come from $TiO_2$ grown in situ on $Ti_3CN$ MXene.

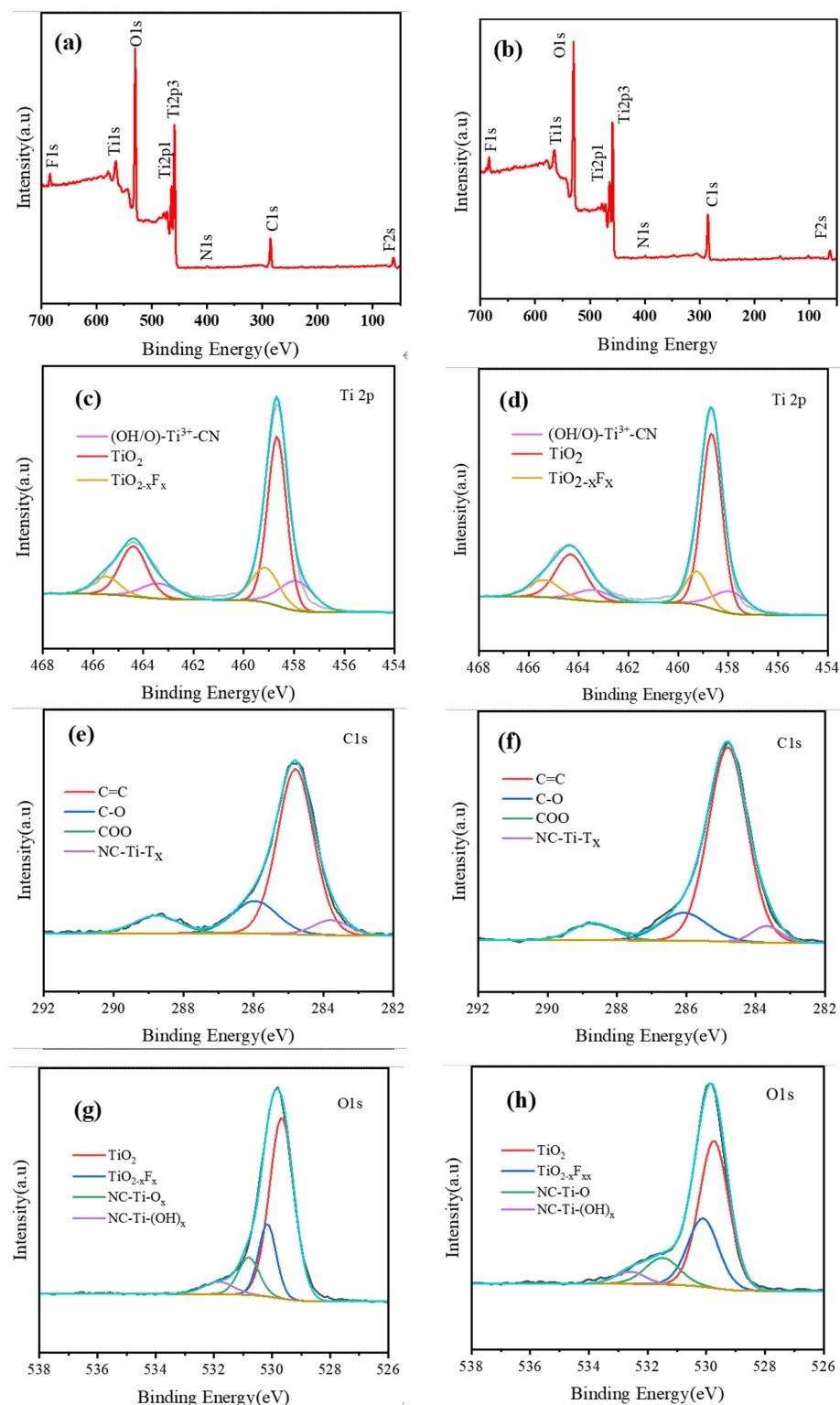

**Figure 8.** XPS full and high-resolution spectra: (**a**) mTOCN full spectrum; (**b**) dTOCN; (**c**) mTOCN Ti2p; (**d**) dTOCN Ti2p; (**e**) mTOCN C1s; (**f**) dTOCN C1s; (**g**) mTOCN O1s; and (**h**) dTOCN O1s.

The high-resolution XPS spectra of Ti2p can be deconvoluted into three peaks [18]: connected with C and surface end-groups -O, -OH to form (OH/O-Ti$^{3+}$-CN) [28], which belongs to Ti$_3$CN(OH)$_x$ connected with C and surface end-groups -F to form (F-Ti-CN), which belongs to Ti$_3$CNF$_x$. The rest is TiO$_2$. This indicates the chemical composition of MXene-based heterojunctions before and after layering treatment is basically the same. The

binding energy of the Ti2p3/2 peak of dTOCN (458.4 eV) shows a slight decrease compared to that of mTOCN (458.7 eV), which is attributed to the connection of more capping groups on the single/fewer layers of $Ti_3CN$, which enhances the electron density in the vicinity of the Ti atom [29].

Figure 8 shows the high-resolution XPS energy spectrum of C1s, which can be deconvolved into four peaks [30], which are $NC-Ti-T_X$ located at by corresponding to 283.78 eV, C=C located at 284.8 eV, C-O at 285.96 eV, and COO at 288.7 eV.

The XPS energy spectrum of O1s could be deconvoluted into three peaks corresponding to the O-C bond as well as the O-Ti bond, where the O-C bond was located at 530.7 eV of the XPS energy spectrum and the O-Ti bond in the anatase $TiO_2$ grown in situ on $Ti_3CN$ after oxidation of the $TiO_2$ was located at 529.8 eV of the XPS energy spectrum [24,30].

### 2.3.2. Band Structure and Photoelectrochemical Characterization

To analyze the energy band structure of the catalyst, the valence band potential (VB) was firstly obtained based on the XPS valence band spectroscopy test, as was shown in Figure 9 XPS valence band spectral curves of the samples dTOCN and mTOCN. The instrumental figure of merit was 4.3 eV, and the valence band potentials of dTOCN and mTOCN were obtained to be 2.4 and 2.29 eV relative to the standard hydrogen electrode, respectively.

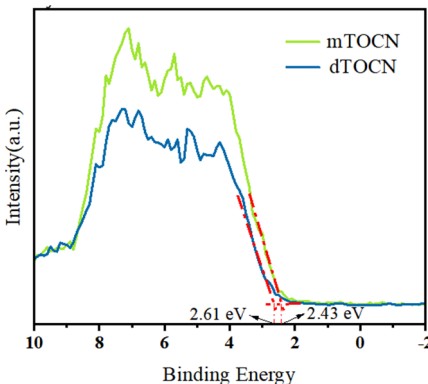

**Figure 9.** XPS valence band spectrum of mTOCN and dTOCN.

The light absorption properties of mTOCN and dTOCN were measured by UV-Vis diffuse reflectance spectroscopy (DRS). From Figure 10, it can be seen the absorption edge of mTOCN is at 325 nm and the absorption region is limited to 400 nm, while the absorption edge of dTOCN undergoes a slight red shift to 350 nm and the absorption region is limited to 410 nm. The Visible light absorption of both heterojunction samples is elevated to some extent at 325–400 nm and 350–410 nm, which indicates the successful configuration of the composite energy bands of the heterojunction samples, whose absorption range of light is expanded from the near-ultraviolet to a part of the Visible range.

After fitting and analyzing the obtained UV diffuse reflectance test spectra, the apparent bandgap of the heterojunction samples can be calculated to further analyze the catalyst energy band structure. The UV spectral data were fitted according to Tauc's formula with n taken as 1/2, and the results were shown in Figure 11. The band gap of sample dTOCN was obtained by making a tangent line of 3.03 eV and that of sample mTOCN was 3.12 eV. The band gap of $Ti_3CN/TiO_2$ was narrower compared to that of the nano-sheets of $TiO_2$ (3.22 eV), which suggested the presence of the metallic $Ti_3CN$ was favorable for light absorption and photocatalytic reaction. The two heterojunctions of dTOCN exhibit higher light absorption and have better photovoltaic performance. The conduction band potential (CB) can be obtained by combining the band gap obtained from the previous spectra by UV diffuse reflectance testing, which are −0.56 and −0.83 eV, respectively.

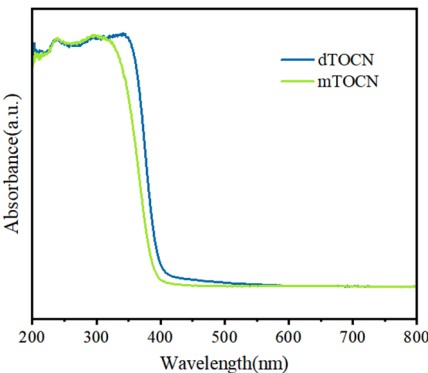

**Figure 10.** UV diffuse reflectance test spectra of dTOCN and mTOCN.

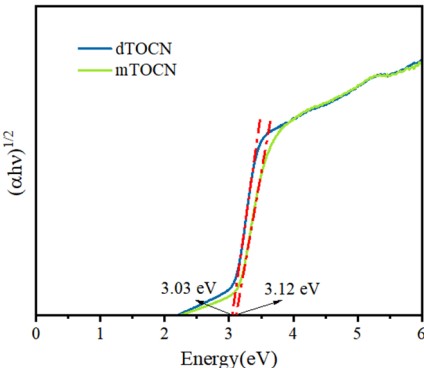

**Figure 11.** Tauc diagram of dTOCN and mTOCN.

### 2.3.3. Photocurrent Response Analysis

Figure 11 shows the photocurrent response plots obtained from the photocurrent tests performed on the samples dTOCN and mTOCN at a constant potential. The photocurrent test results are shown in Figure 12. The photocurrent density of dTOCN is much higher than that of mTOCN, indicating the photogenerated carriers inside the former are generated and migrated more efficiently. This is due to the fact that carrier transport between interfaces is affected by the interfacial structure, and the layered $Ti_3CN$ is closer to the two-dimensional structure and possesses a shorter transport distance, which can effectively inhibit the compounding of the photogenerated carriers and realize the efficient interfacial charge transfer on dTOCN.

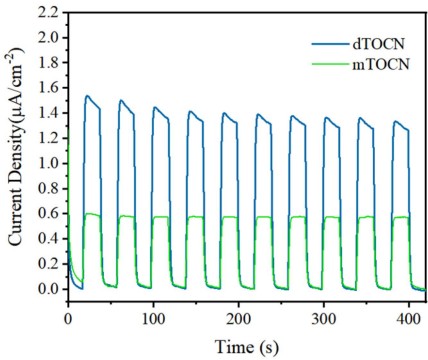

**Figure 12.** Transient photocurrent response graph of dTOCN and mTOCN.

The Nyquist plot of different heterojunctions is shown in Figure 13. The semicircle diameter of the dTOCN curve is smaller than that of mTOCN, indicating the charge transfer resistance of heterojunction materials based on monolayer $Ti_3CN$ is lower than that of

heterostructures based on multilayer Ti$_3$CN, and the higher the conductivity of dTOCN is, the more favorable it is for the transport of photogenerated carriers in the catalyst.

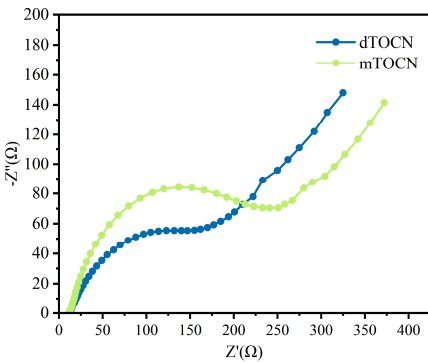

**Figure 13.** Nyquist plot of dTOCN and mTOCN.

Figure 14 shows PL emission spectra of dTOCN and mTOCN. The lower the intensity of the fluorescence emission peak, the lower the carrier recombination rate. The results show dTOCN has higher photogenerated carrier separation and mobility than mTOCN. It is confirmed ultrasonic delamination of MXene can improve the lifetime of photogenerated carriers in heterojunctions.

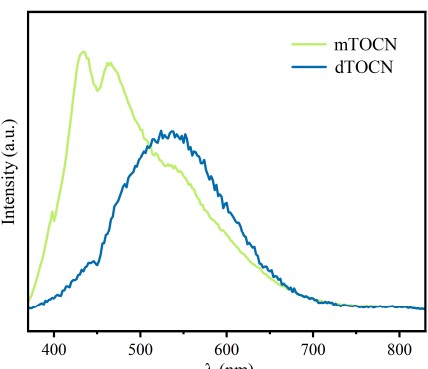

**Figure 14.** Photoluminescence Spectroscopy of dTOCN and mTOCN.

## 3. Materials and Methods

### 3.1. Catalyst Preparation

#### 3.1.1. Multi-Layer and Single-Layer Ti$_3$CN MXene

Firstly, the etchant was prepared by adding 1.6 g of LiF as well as 20 mL of 9 M HCl solution to a PTFE reactor, and stirring magnetically for 15 min, until LiF was completely dissolved, and the solution was light yellow. A total of 1 g Ti$_3$AlCN was measured, and the powder was slowly added to the above etchant. And the reactor was introduced into a 40 °C water bath with sealed magnetic stirring. After 36 h of reaction, the acidic mixture was added to a centrifuge tube and washed with deionized water (centrifuged at 10,000 rpm, 5 min) until pH > 5. Freeze-drying yielded a multilayered Ti$_3$CN MXene sample, noted as mTCN.

To the centrifuged precipitate, 500 mL of deionized water was added, sealed, and N$_2$ bubbled in for 30 min, and sonicated in an ice-water bath under the protection of an inert gas atmosphere for 3 h. The suspension was centrifuged at 3500 rpm for 1 h to remove the unpeeled multilayer of Ti$_3$CN, and the material in the supernatant was exfoliated monolayer of Ti$_3$CN nanosheets. The centrifuged liquid was freeze-dried to obtain a sample of the monolayer of Ti$_3$CN MXene, which was denoted as dTCN.

3.1.2. Ti$_3$CN/TiO$_2$ Heterojunction

A total of 0.2 g of Ti$_3$CN powder was weighed, to which was added 60 mL of 1 M HCl mixed with 1 M NaBF$_4$ solution, then sonicate for 30 min. The mixture was transferred to a 200 mL high-pressure reactor oil bath at 180 °C for 4 h, then cooled at room temperature before the sample was washed with deionized water until nearly neutral, and freeze-dried for 12 h. The Ti$_3$CN/TiO$_2$ heterojunction samples with in situ growth of TiO$_2$ on the surface were obtained, the heterojunctions m with multilayer Ti$_3$CN as the substrate were named mTOCN, and the heterojunctions with monolayer Ti$_3$CN as the substrate were named dTOCN.

*3.2. Characterization Analysis Methods*

The X-ray diffraction (XRD) patterns of the catalyst samples were obtained by an X-ray powder diffractometer (equipment and instrument manufacturer: Bruker, Germany; model: BRUCKER D8 ADVANCE X-ray powder diffractometer) with a generator with a maximum power of 3 kW, a tube voltage of 20–60 kV (1 kV/1 step), a tube current of 10–60 mA, and an accuracy of angular reproducibility of ±0.0001°. The measurement accuracy is ±0.0001°. In this experiment, 2°/min was used, and the scanning angle range was 5–80° to study the crystal shape, physical phase, and crystallinity of the catalyst.

X-ray photoelectron spectroscopy (XPS) is obtained by an X-ray photoelectron spectrometer (equipment and instrument manufacturer: ThermoFisher; model: Thermo escalab 250Xi), which analyzes the depth of analysis up to 10 nm and has a detection limit of 0.01~1at% for elements. This was used to detect the catalyst surface chemical composition, bonding state, valence band spectrum, and content information.

The catalyst morphology and microstructure characterization in this experiment were obtained by the following two characterizations. SEM images of the catalyst samples were obtained by field emission scanning electron microscope (Model: Hitachi, SU8020). The samples Ti$_3$CN, Ti$_3$CN/TiO$_2$ heterojunction were observed to study the micro-morphology of the catalyst. The magnification in the test was 200,000 times, the accelerating voltage was 30 kV, and the spot resolution was 1.4 nm.

The optical and photoelectrochemical properties characterization of the catalysts in this experiment were obtained by the following two characterizations. The UV-Visible absorption spectra (UV-Vis diffuse reflectance spectroscopy DRS) of the catalyst samples were used to detect the full-wavelength light absorption capacity of the catalysts using a UV-Visible-near-infrared spectrophotometer (equipment and instrument manufacturer: Shimadzu Corporation, Japan; model: UV-3600 UV-Visible-near-infrared spectrophotometer). Photoluminescence Spectrometer (FLS1000) were used to obtain the photoluminescence (PL) spectra.

The electrochemical impedance spectroscopy (EIS) and the transient photocurrent response was obtained from an electrochemical workstation (equipment and instrument manufacturer: China Chenhua; model: CHI-760E) with a potential accuracy of ±1 mV, 0.01% of full scale, a constant potential range of ±10 V, a current range of ±10~±0.25 A, an AC impedance frequency range of 0.00001 to 1 MHz, and a current accuracy of 0.2% for the detection of catalysts photogenerated carrier production and the complexation of electron pairs and holes under sample illumination conditions.

*3.3. Photothermal Catalyzed CO$_2$ Reduction Experiment*

The experiments were conducted using a photothermal co-catalytic CO$_2$ reduction reaction system for the evaluation of heterojunction catalytic activity testing, which was equipped with a 300 W Xe light source with full reflector to simulate full-spectrum sunlight, an intermittent reaction system, a customized transparent quartz glass reactor selected for the reactor, a gas chromatograph for gas product detection, along with a vacuum pump for the evacuation operation, and a cooling water circulating pump to control the temperature of cooling water.

The photothermal catalyzed $CO_2$ reduction test was carried out in a 400 mL homemade quartz glass reactor, as shown in Figure 15. Firstly, the height of the Xe light source from the catalyst surface and the current size were adjusted, and the light intensity on the catalyst surface was set to 600 mW/cm$^2$, and the filter used was an AM1.5 filter (full solar spectrum light). The synthesized $Ti_3CN/TiO_2$ heterojunction sample was dispersed in anhydrous ethanol and then uniformly applied on the surface of the light-transmitting glassware, and then dried at 60 °C before placing in the custom-made transparent quartz glass reactor, and the temperature was maintained at 30 °C. The temperature inside the reactor was maintained at 30 °C by turning on the cooling water circulation pump. A mixture of $CO_2$ and $H_2O$ was introduced into the customized transparent quartz glass reactor as the reactants by means of bubble inlet, and then both sides of the reactor were opened, and the vacuum pump was used to repeatedly evacuate and purge the reactor five times with high-purity $CO_2$ gas through the gas-washing bottle containing deionized water, and then after a period of time, the reactor was filled with the mixture of $CO_2$ and $H_2O$. Then, both sides of the reactor and the rotary ports were closed, and the reactor was placed in an airtight condition. The reactor was closed at both sides to make it airtight. The gas in the transparent quartz glass reactor was extracted once with a sampling needle as a blank control group and then the light was turned on, after which the gas products on the surface of the glassware were extracted with a sampling needle every hour and passed into the gas chromatograph for detection, and the photothermal synergistic transformation of CO was carried out for a total of 4 h. The gas mixture was purged five times with ionized water, and then the reactor was filled with the mixture of $CO_2$ and $H_2O$ after a period of time.

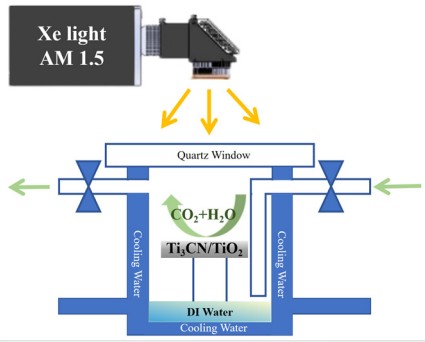

**Figure 15.** Experimental setup with the photothermal catalytic reactor.

## 4. Conclusions

In this paper, $TiO_2$ is generated in situ on $Ti_3CN$ to construct MXene/semiconductor heterojunction. The results of the photothermal catalytic activity test for the photothermal synergistic transformation of $CO_2$ of $Ti_3CN/TiO_2$ heterogeneous constructed on $Ti_3CN$ before and after layering, combined with SEM, XRD, XPS, and UV-Vis DRS, revealed the MXene with low number of layers and small dimensions was closer to the two-dimensional structure. The ultra-thin structure and exposure of more surface-active sites provide a better platform for $TiO_2$ growth, so the $Ti_3CN/TiO_2$ heterojunction prepared by monolayer $Ti_3CN$ has a tighter interface connection, which promotes the rapid migration of carriers between interfaces. The product in the photothermal synergistic transformation of $CO_2$ is CO, and the mTOCN yield constructed from multilayer $Ti_3CN$ is 9.40 µmol·g$^{-1}$·h$^{-1}$, while the dTOCN yield constructed from monolayer $Ti_3CN$ is 11.36 µmol·g$^{-1}$·h$^{-1}$, the catalytic activity is 1.2 times higher than that before delamination of MXene substrate. It is proved the excellent carrier transport capacity is brought by the monolayer $Ti_3CN$ MXene substrate structure, this design can effectively separate photogenerated carriers and improve the reaction rate of photothermal catalytic reduction of $CO_2$. The photocatalytic yield of dTOCN was 7.51 µmol·g$^{-1}$·h$^{-1}$, and the photothermal catalytic effect was enhanced by 1.6 times compared with that of photocatalysis, which showed the excellent photothermal

synergistic effect of $Ti_3CN/TiO_2$ as well as the prospect of monolayer $Ti_3CN$ as an excellent co-catalyst for photothermal catalysis.

**Author Contributions:** Conceptualization, M.G., Z.Y. and Z.W. methodology, Y.Y., J.R., Z.W., J.Q. and J.H.; software, J.Q. and C.Z.; validation, Z.W., J.Q. and C.Z.; formal analysis, J.H. and C.Z.; investigation, C.Z. and Z.W.; resources, M.G., Z.Y., Y.Y. and J.R.; data curation, C.Z. and Y.G.; writing—original draft preparation, C.Z.; writing—review and editing, Z.Y. and M.G.; supervision, Z.Y.; project administration, M.G.; funding acquisition, M.G. All authors have read and agreed to the published version of the manuscript.

**Funding:** This research was funded by National Natural Science Foundation of China, grant number 52276099; Graduate Research and Innovation Foundation of Chongqing, China, grant number CYB23034.

**Data Availability Statement:** Data sharing is not applicable to this article.

**Conflicts of Interest:** The authors declare no conflicts of interest.

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
