# Peer review of "Enhancement of Carrier Migration by Monolayer MXene Structure in Ti3CN/TiO2 Heterojunction to Achieve Efficient Photothermal Synergistic Transformation of CO2"

_catalysts, doi:10.3390/catal14010035_

Round 1

Reviewer 1 Report

Comments and Suggestions for Authors

1. The introduction looks incomplete. It is not clear from it why such a heterostructure was chosen for further analysis. There is no review of heterostructures based on MX`enes. In particular, the introduction should refer to a recent paper (DOI 10.3390/app13169410), which also discusses the creation of heterostructures based on maxene for photocatalytic transformation of carbon dioxide.
2. When discussing the XRD data, I suggest using the method proposed in (DOI 10.3390/ma15248798) to refine the unit cell parameters of MX`ene.
3. When discussing the scheme of photocatalytic reactions, I strongly recommend to add its schematic representation as a separate figure. It will simplify understanding of the essence of the process.

Comments on the Quality of English Language

The quality of English should be improved. There is use of terms that can only be explained by machine translation. So in line 173 we see "incident light" instead of "initiating light", in line 265 "forbidden bandwidth" instead of the accepted designation of the width of the forbidden zone as "bandgap" and so on.

Reviewer 2 Report

Comments and Suggestions for Authors

In this article, C. Zhu et.al., reported the transformation of CO2 using monolayer MXene structure in Ti3CN/TiO2 heterojunction by utilizing the efficient synergistic photothermal. The authors have done good analysis and the manuscript is well organized. The concept theme will definitely grab the researcher’s attention in the filed of 2D based hybrid heterostructure materials in CO2 conversion/reduction applications. However, there are some minor concerns that needs to be clarified before the publication, and here are some of my comments as given below.

1.      Please mention the facets of the XRD patterns for better understanding to the readers.

2.      Importance of 2D materials and layered like nanostructure should be highlighted in the introduction. Because there are plenty of 2D based materials has been used CO2 conversion. So, it may be led to improve the novelty of your study.  

3.      Importance of making heterostructure Ti3CN/TiO2 heterojunction should be highlight and its significance (benefits) of electron charge transfer under photothermal reactions are to be included revised version of manuscript.

4.      Detailed information of the formation of TiO2 nanostructures were missed. Please enlighten this factor for better understanding.

5.      Significance of the photothermal and photo+thermal and their synergistic effects should further add in the introduction in comparison with the photoelectrochemical CO2 conversion.

6.      Please proof read the entire manuscript and check the minor grammatical errors. Please crosscheck and follow the Catalyst Journal manuscript format.

7.      Please provide the impendence spectra and PL analysis for better understanding of the charge carrier transfer phenomenon and charge carrier migration dynamics.

8.      How about the long-term stability of the as-prepared heterostructure catalysts?

9.      Some of the references are not up to the mark, a few of the latest reports are suggested to cite mechanism and 2D materials related discussions, such as Chinese Journal of Catalysis, 47, 243-253, ACS Materials Lett. 2023, 5, 10, 2739–2746, J. Mater. Chem. A, 2023,11, 4230-4237, Chemical Engineering Journal 452 (2023): 139392.

Comments on the Quality of English Language

There are some minor grammatical errors that can be rectified if authors can do the quick proof read.

Round 2

Reviewer 1 Report

Comments and Suggestions for Authors

I believe that the article can be accepted for publication in its current form.